# The eyes don't have it: Eye movements are unlikely to reflect refreshing in working memory

Vanessa M. Loaiza[1]*, Alessandra S. Souza[2,3]

1 Department of Psychology, University of Essex, Colchester, United Kingdom, 2 Faculty of Psychology and Education Sciences, University of Porto, Porto, Portugal, 3 Department of Psychology, University of Zurich, Zurich, Switzerland

* v.loaiza@essex.ac.uk

**Data Availability Statement:** The pre-registration for Experiment 1 and the experimental materials, raw data, and analysis scripts for all experiments are available at https://osf.io/qkmtc/.

## Abstract

There is a growing interest in specifying the mechanisms underlying refreshing, i.e., the use of attention to keep working memory (WM) contents accessible. Here, we examined whether participants' visual fixations during the retention interval of a WM task indicate the current focus of internal attention, thereby serving as an online measure of refreshing. Eye movements were recorded while participants studied and maintained an array of colored dots followed by probed recall of one (Experiments 1A and 1B) or all (Experiment 2) of the memoranda via a continuous color wheel. Experiments 1A and 2 entailed an unfilled retention interval in which refreshing is assumed to occur spontaneously, and Experiment 1B entailed a retention interval embedded with cues prompting the sequential refreshment of a subset of the memoranda. During the retention interval, fixations revisited the locations occupied by the memoranda, consistent with a looking-at-nothing phenomenon in WM, but the pattern was only evident when placeholders were onscreen in Experiment 2, indicating that most of these fixations may largely reflect random gaze. Furthermore, spontaneous fixations did not predict recall precision (Experiments 1A and 2), even when ensuring that they did not reflect random gaze (Experiment 2). In Experiment 1B, refreshing cues increased fixations to the eventually tested target and predicted better recall precision, which interacted with an overall benefit of target fixations, such that the benefit of fixations decreased as the number of refreshing cues increased. Thus, fixations under spontaneous conditions had no credible effect on recall precision, whereas the beneficial effect of fixations under instructed refreshing conditions may indicate situations in which cues were disregarded. Consequently, we conclude that eye movements do not seem suitable as an online measure of refreshing.

## Introduction

Working memory (WM) is the system that briefly holds and manipulates information in mind from moment-to-moment. Much work concerns the role of attention to prioritize the most

**Funding:** Data collection of Experiment 2 was support by a grant from the Swiss National Science Foundation to A. S. Souza (project 100019_169302). The funders had no role in the study design, data collection and analysis, decision to publish, or preparation of the manuscript.

**Competing interests:** The authors have declared that no competing interests exist.

relevant contents of WM via *refreshing* [see 1 for a recent review]. Refreshing is considered a domain-general function that brings a representation into the focus of attention in WM, thereby improving its accessibility. A persistent quest is to find direct evidence that refreshing exists and has a functional role in attention-based maintenance in WM.

One method of manipulating refreshing is to explicitly instruct participants to refresh [2–6]. For example, in their instructed-refreshing paradigm, Souza and colleagues presented a memory array (e.g., to-be-remembered colors) with one of the items later probed. During the retention interval (RI), a series of sequentially presented cues (i.e., arrows) pointed to the memory items between 0 and 2 times to prompt participants to "think of" (i.e., refresh) the cued items. A beneficial impact of refreshing was observed: items cued more often to be refreshed were better recalled from WM [4, 5], regardless of their modality (verbal or visuospatial) [7].

This paradigm reveals that instructed refreshing benefits WM. However, it does not elucidate how participants use the cues or whether this underlying process is similar to what participants spontaneously do in a typical WM paradigm without instruction. Furthermore, there is no means to measure whether all the cues were followed or how much attention they engaged. Thus, there may be substantial individual differences in instruction-following, and hence, in in the cues' facilitative effect. In this paradigm, the only evidence that the cues were used was their beneficial effect on recall. If the cues did not help, it would be impossible to disentangle whether refreshing is ineffective, or participants simply ignored the cues.

One means to address these issues is to consider an online yet unobtrusive measure, such as eye movements, of when and where participants focus their attention. Much work has demonstrated the link between eye movements and covert shifts of attention, especially the *looking-at-nothing phenomenon* [8–10]. It is often found that fixating at now-empty locations of previously presented memoranda facilitates retrieval from episodic memory [11, 12]. This supports the notion that covert shifts of attention, as reflected by fixations, have a functional purpose, perhaps because looking at the now-empty location reactivates the associated content [8].

Curiously, the looking-at-nothing phenomenon has been largely overlooked by researchers interested in refreshing despite the fact that the notion of reactivation through covert shifts of attention greatly resonates between the literatures. One recent exception investigating eye movements in visual WM has shown that participants spontaneously gazed toward the location of a tested item despite no instruction or incentive to do so [13]. Furthermore, presenting a cue after encoding to indicate which item would be tested (i.e., retro-cue) biased gaze toward the location of the retro-cued item, and the size of the gaze bias predicted response times during WM recall. These results cohere with those of the looking-at-nothing literature by suggesting that fixations may reflect covert shifts of attention. Still, it remains unclear whether fixations on now-empty locations of previously presented memoranda are functional in WM and hence could be linked to acts of refreshing, and whether spontaneous and instructed refreshing as measured by fixations yield similar WM improvements.

In the current study, we investigated whether fixations indicate acts of refreshing in a classic visual WM task. If so, then we can make a more explicit link between spontaneous and instructed refreshing using fixations as an online measure of refreshing in WM. Participants maintained a memory array of colored dots for a probed recall test of one (Experiments 1A and 1B) or all of the dots (Experiment 2) along a continuous color wheel (Fig 1). The measure of performance was the distance between the tested and the reported color (i.e., recall error). Crucial to our question are the events during the RI. In Experiment 1A, participants simply maintained the colors during an unfilled 2.5-s or 4-s RI with no further instruction. In Experiment 1B, the same group of participants was instructed to "think of" a cued color 0, 1, or 2 times during the RI according to a series of refreshing cues (arrows) presented for either 0.5s

A. E1A: Baseline/Unfilled RI  B. E1B: Instructed Refreshing C. E2: Unfilled RI with/without placeholders

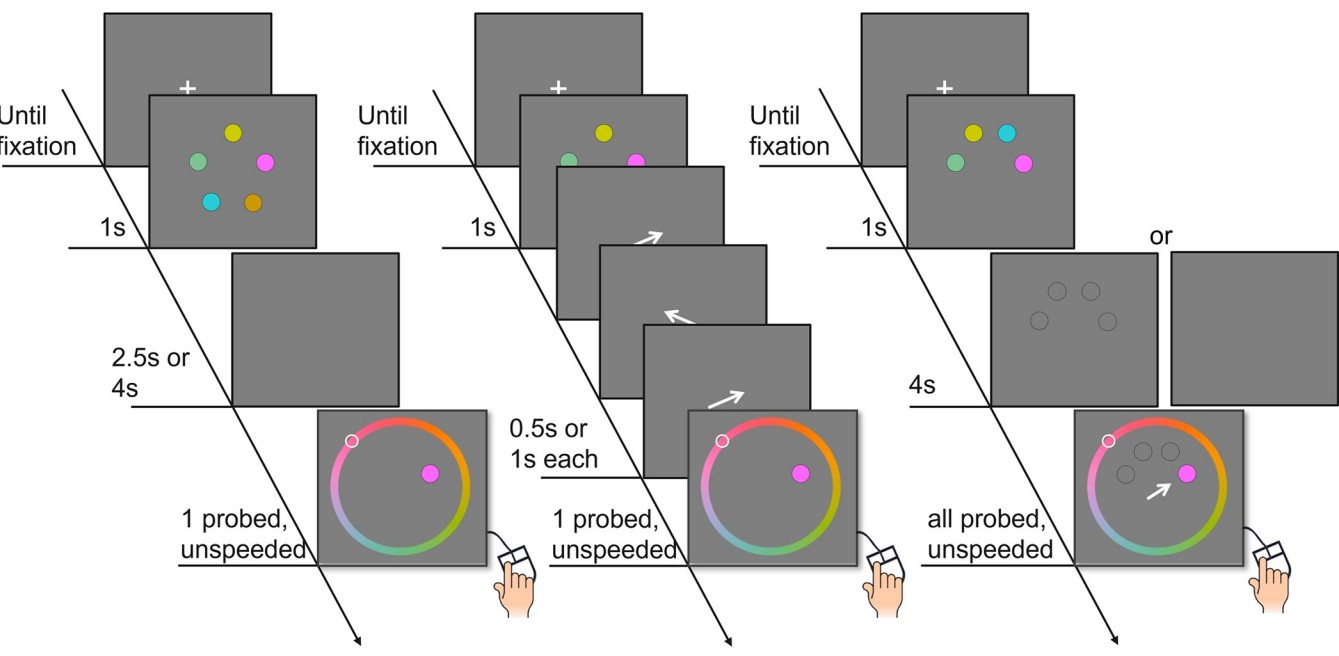

*Note.* E = experiment, RI = retention interval. See the online article for the color version of this figure.

**Fig 1.** Example of the Task in Experiments 1A (Panel A), 1B (Panel B), and 2 (Panel C).

or 1s thereby varying cue frequency and duration, respectively. Thus, the aim of Experiments 1A and 1B were to investigate whether fixations indicate acts of refreshing in spontaneous and instructed conditions, respectively. As will become clearer further on, the results of Experiment 1B were relatively conclusive whereas those of Experiment 1A were not, and so we designed Experiment 2 to specifically address the lingering issues of Experiment 1A regarding fixations under spontaneous conditions. Specifically, during Experiment 2, a semi-circular array of 4 colored dots was displayed followed by a RI wherein either the screen remained blank, or placeholders of the to-be-remembered dots remained on screen. The design of these experiments allowed us to investigate three main research questions.

First, we considered the occurrence of fixations toward previously presented locations of the memoranda during the RI (i.e., looking-at-nothing). We assessed this by considering fixation rate (i.e., fixations per second) to the memoranda compared to other screen locations during encoding and the RI. Fixations during encoding are often associated with heightened attention to these items gating their preferential encoding [14–16]. If attention continues to be engaged to the memoranda during maintenance, then fixations should be directed more often to memory locations during the RI as well [13, 17].

Given that the memoranda were evenly distributed across the array in Experiment 1A, it was difficult to clearly distinguish whether fixations in any direction reflected either looking-at-nothing or just random looking. Thus, Experiment 2 more reliably associated fixations with a looking-at-nothing strategy by presenting memoranda in a randomly determined half of the screen. This allowed us to assess how often participants looked back to locations of previous memoranda versus previously empty locations. We also varied the presence of placeholders during retention to consider whether this could facilitate gaze-based rehearsal [18, 19]. If participants are biased toward looking at the locations of currently maintained items in memory,

we should observe far more fixations toward locations of previously presented memoranda compared to other locations.

Second, in Experiments 1A and 2 we examined whether spontaneous looking-at-nothing behavior during a blank RI predicts WM recall. If fixations reflect functional shifts of attention for refreshing, then increasing fixations toward a now-empty item location should improve its recall when this item is eventually tested.

Third, we examined whether fixations toward a cued location index the attention allocated to the cued item, thereby providing an independent predictor of cue use. Thus, in Experiment 1B, we recorded how fixations changed in relation to the presentation of refreshing cues during the RI. Consistent with previous work [4, 7], recall should improve with increasing refreshing cues. Critically, if fixations provide an online and unobtrusive measure of attentional allocation, then they should capture an additional source of individual variation in the use of the cues to guide attention to memoranda.

Altogether, answering these three questions will indicate if fixations can provide a useful tool to study refreshing online, both when it occurs spontaneously as well as when instructed by the experimenter.

## General method

### Participants

In Experiments 1A and 1B, we collected data from 30 adults ($M_{age}$ = 20.54, $SD$ = 1.82) from the participant pool at the University of Essex who were compensated with partial course credit or £15. An additional five participants were excluded from analysis due to failure to complete the experiment (e.g., failing to pass the first eye calibration procedure after multiple attempts; $n$ = 3) or due to experimenter error causing the loss of the eye-tracking data ($n$ = 2). Sample size was determined based on our previous experience with these tasks and effects. In line with our pre-registration (see https://osf.io/vqd76/), we monitored the evidence for our hypotheses after the initial 30 datasets were available, with the plan to continue collecting to a maximum of 50 participants (due to resource limitations) if the evidence for the effects of interest up to that point was ambiguous. In the pre-registration of Experiment 1, we had assumed that we would use Bayes factors to decide whether the results were ambiguous. Given that we used a mixed effects approach described further on, however, we used credible intervals instead to draw inferences. Relying on Bayesian inference allows our analyses to be unbiased by changes in sampling plan after looking at the data [20].

In Experiment 2, we collected data from 19 adults ($M_{age}$ = 22.95, SD = 2.76) from the participant pool at the University of Zurich who were compensated with partial course credit or 15 CHF. The data of one additional participant were excluded from analysis due to experiment failure. Experiment 2 was not pre-registered, so there is no sample size justification; our aim was to recruit a similar number of participants as Experiment 1. Due to the coronavirus pandemic, we were not able to continue with data collection.

All participants provided written informed consent and were debriefed at the conclusion of the experiments. Experiment 1 was approved by the ethics committee at the University of Essex and Experiment 2 was conducted in accordance with University of Zurich regulations (automatic approval via an ethic self-check list). All participants self-reported normal or corrected-to-normal vision and normal color vision.

### Materials

The materials for both experiments can be found on the Open Science Framework (OSF): https://osf.io/qkmtc/. Experiments 1A and 1B were programmed in Matlab using the

Psychophysics and Eyelink Toolbox extensions [21–23; see http://psychtoolbox.org]. These experiments were presented on a 50x30 cm CRT monitor with resolution of 1920x1080 with 75% illumination. Participants were seated at a distance of 80 cm from the computer screen and their heads were supported by a chinrest. Eye movements of the right eye were recorded using a desk-based SR Research Eyelink CL 1000/2000 eye-tracker sampling eye position at 1000 Hz. The eye-tracker was calibrated using a nine-point calibration procedure, which occurred before receiving instructions for and approximately every 20 trials during the main tasks. Following successful calibration of the eye-tracker, participants received instructions for or continued with the task.

In Experiment 2, the eye-tracker comprised a GazePoint 150 Hz device, which was connected with Matlab via the iMotions software. The experiment was presented on LCD 56.8 x 33.5 cm BenQ XL2430T monitor with a resolution of 1920x1080 and refreshing rate of 144 Hz. The eye-tracker was connected to the same computer that presented the stimuli. Participants sat at distance of 60 cm from the computer screen with their heads supported by a chinrest. In Experiment 2, eye movements of both eyes were tracked with a sampling rate of 150 Hz. The eye-tracker was calibrated with a nine-point calibration procedure in the beginning of the experiment only.

The main task comprised remembering the colors of a set of dots, and to reproduce the color of a probed item using a continuous color wheel. To-be-remembered dots (dots radius = 32 pixels) were sampled randomly from a continuous color wheel of 360 possible colors from a circle in the CIELAB color space (L = 70, a = 20, b = 38, radius = 60). In Experiments 1A and 1B, the memory set consisted of a variable number of dots that were evenly spaced on an imaginary circle (radius = 200 pixels). In Experiment 2, the memory set consisted of four dots which were presented in four out of eight locations evenly spaced along an imaginary circle (radius = 250 pixels). The four locations were selected such that half of the screen was occupied with memoranda.

## Procedure

**Experiment 1A and 1B.** Participants were tested individually in quiet booths with an experimenter present to ensure that the instructions were understood and followed and to ensure proper functioning of the eye-tracker. After completing a web-based color vision test, participants completed a set size calibration phase without the eye-tracker, followed by two sessions of a visual WM task with the eye-tracker: Experiment 1A, serving as the baseline session with no refreshing cues during the RI, and Experiment 1B in which refreshing cues were presented during the RI of the task.

During the set size calibration task, participants completed 40 trials (with four practice trials) of a visual WM task wherein the set size (i.e., the number of presented to-be-remembered colored dots) was gradually adjusted to achieve a criterion level of 40° of recall error. To begin each trial, a fixation cross was displayed for 0.5s followed by a set of $n$ colored dots presented for 1s. After a brief RI of 2.5s or 4s (half of either length, randomly intermixed), memory for one of the colored dots was tested by presenting a dark-grey disk at the location of one of the dots (i.e., probe) surrounded by a color wheel around the location of all the dots and a mouse cursor at the center of the screen. Participants moved the mouse around the color wheel to adjust the color of the probe disk to the color they remembered as being presented in that location. When they were satisfied with their answer, they pressed the left-mouse button to confirm their response, and a new trial began after a 1.5s inter-trial interval (ITI). The initial set size of the memory array was 6 items, and based on participants' ongoing performance in the last 4 trials, the $n$ in the follow trial was adjusted. If mean recall error fell below 40°, then $n$ was

increased by 1; if it exceeded 40˚, then $n$ was decreased by 1. Set size could range between 3 and maximum of 10 colored dots. The average $n$ in the last 20 trials was used to determine the set size used in Experiments 1A and 1B. For example, if the average $n$ = 5.3 items, then 70% of the trials in Experiments 1A and 1B contained a set size of 5 items and 30% of the trials contained 6 items, randomly intermixed. The calibration phase was successful at adapting participants' recall error close to 40˚ ($M$ = 44.66, $SD$ = 5.87) with a set size of about 5–6 items for most participants ($M$ = 5.76, $SD$ = 1.00, range = 4.15–8.40), similar to our previous work [24].

Participants next completed the baseline Experiment 1A. There were four practice trials and 100 critical trials (Fig 1A). Participants began each trial by fixating on a fixation cross; the trial started when the eye-tracker detected their fixation on the cross. If fixation was not detected within 5s, the experimenter re-positioned the camera if necessary and reminded the participants to fixate on the cross before re-starting the procedure. If it still failed, then calibration of the eye-tracker was reinstated by the experimenter. Participants were informed that they could freely move their eyes thereafter. The trials then progressed much like the set size calibration phase: a memory array of $n$ colored dots was presented for 1s, with the $n$ having been individually determined during the set size calibration phase. After a brief RI of 2.5s or 4s (approximately half the trials of either length, randomly intermixed), memory for one of the colored dots was tested by presenting a dark-grey probe disk in the original location of one of the dots. Participants moved the mouse cursor from the center of the screen around the continuous color wheel surrounding the location of all the dots to select the color they remembered as having been presented in that location. Note that we aimed to have 50 trials per design cell in Experiments 1A (i.e., RI) and 1B (i.e., cue duration and number of refreshings). However, it was not evident until late into data collection that the trials of both Experiments 1A and 1B were inadvertently unbalanced. Thus, the split of the trials across the designs was only approximately even, but the cell size varied across participants (e.g., 48 vs. 52 between two conditions). Although inconvenient, all of the principal analyses pertaining to the research questions were conducted at the trial level, and thus there is no problem for drawing inferences.

Following completion of Experiment 1A, participants were allowed a break before beginning Experiment 1B. After successfully completing the eye-tracking calibration procedure, participants received instructions to complete four practice trials and 300 critical trials (Fig 1B). As in Experiment 1A, a fixation cross appeared onscreen and required participants' fixation before the trial progressed to the presentation of the memory array of $n$ colored dots for 1s. Following a 0.5s interstimulus interval (ISI), three sequentially presented white arrows (i.e., refreshing cues) appeared at the center of the screen, each pointing to the location of one of the to-be-remembered dots. The refreshing cues were presented for either 0.5s or 1s, with trials of each duration randomly intermixed. Following the offset of the last refreshing cue, a final ISI of 0.5s preceded the test. Thus, the entire RI of Experiment 1B lasted either 2.5 or 4s depending on the cue duration, just as in Experiment 1A. Participants were instructed to think about the color of the dot that each arrow points to for as long as it is presented, with the instructions specifying that this is the most important task in the block. Participants were also explicitly informed that the cues did not predict which item would be tested. There were two possible sequences of the refreshing cues: Assume ABC refers to three randomly selected colors from the memory array, then the refreshing cues could point to three different items (A-B-C) or two different items once and a second item twice (A-B-A). These were the only two possible sequences of refreshing cues; thus, the cues always alternated between different items, and no items were cued twice in succession. Thereafter, participants' memory was tested as in the previous tasks, with a dark-grey probe disk in the location of one of the colors and surrounded by the continuous color wheel.

**Experiment 2.** Participants in Experiment 2 were tested in up to groups of 2 at individual stations, with an experimenter present to answer questions and ensure that instructions were followed. Experiment 2 was similar in procedure to the previous experiments (Fig 1C): After receiving instructions and 12 practice trials, participants completed two blocks of 60 trials, with each trial beginning with a cross requiring the participants' fixation before presenting a memory array of four colored dots for 1s. The fixation cross remained at the center of the screen for the duration of the trial. Four contiguous locations along an imaginary circle of 8 possible fixed spatial locations were selected such that the to-be-remembered dots were randomly clustered on the top, right, or left side of the screen (see Fig 1C). The two blocks of trials only differed regarding whether the memoranda disappeared entirely from the screen during the RI, as in Experiment 1A, or black outlines of the memoranda remained on the screen during the RI. The order of the blocks was counterbalanced across participants. After 4s, each of the memoranda were probed in a random order with an arrow pointing to each dot, prompting the participant to use the mouse to select the color from the continuous color wheel that they remembered for that probed location. Note that, due to experiment failure, eight participants did not fully complete the second block of trials. Once again, all of the primary analyses were conducted at the trial level, and so this issue is inconvenient but not problematic for drawing inferences.

## Data pre-processing and analysis

The raw data and analysis scripts can be found on the OSF: https://osf.io/qkmtc/. The pre-processing of the raw data and analyses were conducted in R [25]. Offline parsing of the raw eye-tracking data was conducted with the R package "saccades" [26] that applies Engbert and Kliegl's [27] velocity-based classifying algorithm. The instances where a timestamp was misprinted in the raw eye-tracking data were removed before parsing (0.00005% and 0.54% of the samples in Experiments 1 and 2, respectively). Practice trials were excluded from parsing and analysis. Although eye movements were recorded and parsed for the full trial sequences, henceforth we focus on the encoding and RI phases. Any events that were not classified as a fixation (i.e., blinks or events that were deemed "too short" or "too dispersed" artifacts) were excluded from analysis (approximately 23.1% and 19.7% of the detected events in Experiments 1 and 2, respectively). Finally, in Experiment 1 there were several instances that occurred where the program crashed and had to be re-started, resuming at the same block and trial where the crash had occurred. In these cases (0.05% of the trials), the trial where the crashed occurred and the one following were removed from analysis.

During data processing, we defined a threshold radius around the area of the dots to qualify looking at the location of a given dot versus the center of the screen or anything else. This threshold was defined as a radius of 142 pixels, which is the radius around 4 dots (i.e., the minimum calibrated set size in Experiment 1 and the presented set size in Experiment 2) before touching the area around the next dot (allowing classification of 76%, 67%, and 59% of the detected fixations in Experiments 1A, 1B, and 2, respectively). We also considered a stricter threshold for both experiments, the analyses for which can be found on the OSF. In the instances where a fixation was ambiguous (i.e., classified as fixating on two dots, 17.2% and 16.4% of fixations in Experiments 1 and 2, respectively), the dot with the minimum distance to the fixation was taken. The threshold for looking at the center of the screen was a radius of 57.5 pixels, which is the minimum radius before touching the aforementioned defined area surrounding the dots (reflecting 16%, 27%, and 23% of the detected fixations in Experiments 1A, 1B, and 2, respectively). Finally, fixations falling outside of these areas were labeled "other" (7%, 6%, and 18% for 1A, 1B and 2, respectively); in Experiment 2, these "other" areas could

include stimulus locations not used during the current trial (e.g., when a trial showed the memoranda on the left, the unused center and right locations were "other" during that trial). In Experiment 1, only 14% of the total fixations were qualified as looking toward the eventually tested item during the RI. The fact that most of the fixations to the memoranda were essentially irrelevant to the eventually tested item prompted us to test all the presented items in Experiment 2 so as to have more data for analysis.

We assessed the evidence for our hypotheses using Bayesian inference. Bayesian inference involves updating one's prior beliefs about some parameters of interest in light of the observed data. For example, key parameters of interest in this work concern the effect of eye movements during the RI on WM performance and its interaction with other manipulated independent variables, such as the number of presented refreshing cues. The updated beliefs are the resulting posterior distributions of each of these parameters that serve as means to assess how confident we are that the parameters are credibly different from 0. For example, one could observe whether the interval covering 95% of the posterior distribution (i.e., the credibility interval, CI) for the effect of fixations includes 0.

To this end, we used the brms package [28] to conduct Bayesian mixed effects models. The mixed effects approach was ideal because it allowed us to account for variability in fixations at both the participant level and the trial level, which was particularly useful for assessing the effect of fixations on WM recall. The brms package uses Stan [29] to estimate posterior distributions of the model's parameter estimates using Monte-Carlo algorithms. An advantage of brms is that it comprises different families of distributions (most relevantly, ex-Gaussian and von Mises) that can be fit to the data, as we explain further on. We applied uninformative/flat priors on the coefficients of the effects and intercept for all the tested models. The posterior parameter estimates of all the tested models were sampled through four independent Markov chains each comprising 2,000 iterations, with the first 1,000 warmup iterations excluded from analysis. We checked the chains for convergence via visual inspection as well as verifying that the $\hat{R}$s of the fitted models' parameters were close to 1. Posterior predictive checks ensured appropriate model fit to the data.

The analysis for our first research question concerned fixation rate (i.e., the number of detected fixations per second) to the memoranda, center, and other screen locations as a function of phase (Encoding vs. RI) and RI duration (Experiment 1; 2.5s vs. 4s) or placeholders (Experiment 2; absent vs. present). Each of these variables was manipulated within-subjects and treated as a fixed effect in the analysis, with the condition listed first in Table 1 serving as the reference. Additionally, fixation rate in Experiment 1B to either the cued locations or eventually tested target was also considered as a function of cue duration (0.5s vs. 1s) and the number of times the cued location or eventually tested item was cued (0, 1, or 2), respectively. For analysis, fixation rate was aggregated across trials for each of these relevant variables for each participant. We also adjusted fixation rate according to the time allotted during the encoding (1s) and RI (2.5s or 4s) phases to ensure that the measure was comparable between these phases. Fixation rate was assumed to follow an ex-Gaussian distribution because it was left-skewed with a long tail (see Fig 2, top panels). The effects of the factors were applied to the mean of the distribution, and their effects on the additional parameters of sigma (i.e., standard deviation of the Gaussian component) and beta (i.e., scale of the exponential component) were not included.

The analyses concerning our second and third research questions concerned recall error as a function of fixation frequency to the eventually tested target(s) during the RI (i.e., the proportion of fixations to the target(s) out of the total detected fixations for each trial) and the independent variables of each experiment. Alternative analyses instead using proportion fixation duration (i.e., the proportion of time spent looking at the target(s) out of the total duration

**Table 1. Research question 1: Mean posterior estimates [and 95% credibility intervals, CIs] of the effects of the predictors on fixation rate in each experiment.** The Condition Appearing First in the Parentheses was the Reference Variable.

| Effect | Experiment 1A | Experiment 1B | Experiment 2 |
|---|---|---|---|
| Intercept | **1.56 [1.46, 1.67]** | **1.72 [1.49, 1.93]** | **1.34 [1.13, 1.57]** |
| Phase (encoding vs. RI) | **-1.00 [-1.11, -0.88]** | -0.14 [-0.34, 0.07] | **-0.50 [0.71, -0.30]** |
| RI (2.5 s vs. 4 s) | 0.01 [-0.07, 0.08] | 0.04 [-0.02, 0.11] | - |
| Placeholders (absent vs. present) | - | - | -0.06 [-0.20, 0.08] |
| Location (center vs. other) | **-1.23 [-1.35, -1.12]** | **-1.48 [-1.65, -1.30]** | **-0.70 [-1.03, -0.38]** |
| Location (center vs. dots) | **4.59 [3.69, 5.52]** | **3.40 [2.65, 4.12]** | **1.59 [0.83, 2.34]** |
| Phase x RI | 0.07 [-0.04, 0.19] | **-0.17 [-0.27, -0.08]** | - |
| Phase x Placeholders | - | - | **-0.27 [-0.47, 0.06]** |
| Phase x Location (center vs. other) | **1.12 [0.98, 1.26]** | **0.35 [0.19, 0.51]** | **0.68 [0.39, 0.97]** |
| Phase x Location (center vs. dots) | **-1.45 [-2.00, -0.90]** | **-1.21 [-1.75, 0.64]** | **-1.11 [-1.54, -0.70]** |
| RI x Location (center vs. other) | 0.01 [-0.09, 0.11] | -0.05 [-0.13, 0.02] | - |
| RI x Location (center vs. dots) | -0.12 [-0.39, 0.15] | **0.23 [0.03, 0.44]** | - |
| Placeholders x Location (center vs. other) | - | - | 0.09 [-0.11, 0.30] |
| Placeholders x Location (center vs. dots) | - | - | 0.09 [-0.12, 0.31] |
| Three-way interaction (center vs. other) | -0.09 [-0.25, 0.06] | **0.16 [0.06, 0.27]** | -0.07 [-0.35, 0.22] |
| Three-way interaction (center vs. dots) | **-0.42 [-0.71, -0.13]** | **-0.62 [-0.83, -0.40]** | **1.06 [0.61, 1.49]** |

*Note.* Boldface font denotes a credible effect (i.e., CIs do not overlap with 0). RI = retention interval. The three-way interaction included the factors of phase, location, and RI (Experiment 1) or placeholders (Experiment 2).

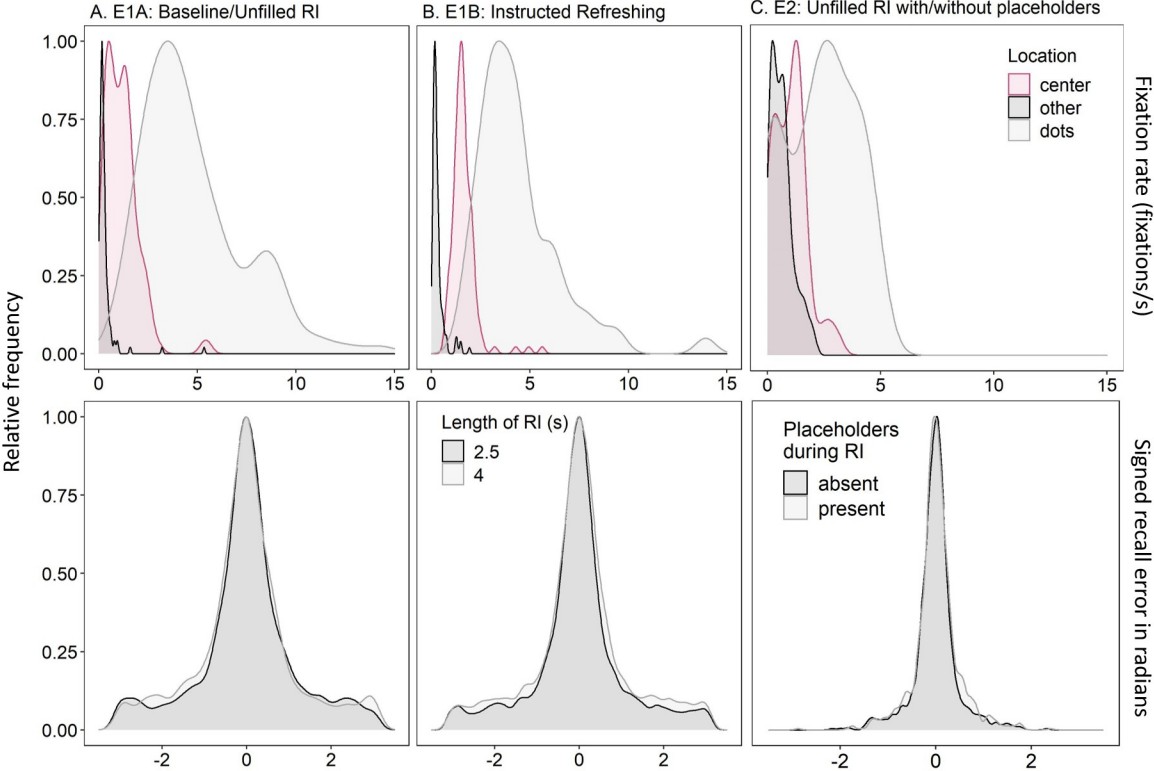

**Fig 2.** Density Plots of Fixation Rate (Fixations Per Second; Top Panels) and Recall Error (Transformed into Radians; Bottom Panels) in Experiments 1A (Panel A), 1B (Panel B), and 2 (Panel C).

of detected fixations) yielded similar results to those reported. This is not surprising given that the measures were highly correlated ($r$s ranging .80 to .92). These alternative analyses can be found on the OSF. The independent variable(s) in Experiment 1A were RI (2.5s or 4s) and in Experiment 1B the number (0, 1, or 2) and duration (0.5s or 1s) of the presented cues to instruct refreshing. Experiment 2 manipulated the presence of placeholders of the memoranda during the RI (absent or present), and we also included an overall effect of test order (i.e., output position, 1 to 4) given that all of the memoranda were tested. Each of these within-subjects variables were entered as a fixed effect in the analysis, with the first level listed above serving as the reference, except the number of presented cues (Experiment 1B) and test order (Experiment 2) which, like fixation frequency during the RI (in all experiments), were treated as continuous predictors. The signed recall error (ranging from -180 to 180˚) was transformed from degrees into radians in order to fit the data with a von Mises distribution, a circular normal distribution. The effects of the variables were applied to the kappa parameter, representing the precision of the von Mises distribution; the mean of the distribution was set to 0 given that responses were centered on the value of the tested item (see Fig 2, bottom panels). Thus, recall precision (i.e., kappa) was the specific outcome variable of interest to address our second and third research questions.

During the review process, we conducted two further alternative analyses that can be found on the OSF: First, we applied the effects of the variables to the mean of the von Mises distribution rather than kappa, but given the clear mean of 0 shown in Fig 2, there were no credible effects on the mean. Second, we fit an ex-Gaussian distribution to recall error in degrees as the dependent variable and the effects of the variables applied to the mean of the distribution. In most instances this model failed to converge (i.e., $\hat{R}$s greatly exceeding 1.06) even with many more iterations per chain, whereas in the other instances where the model converged, the pattern of results was the same as the von Mises distribution. Thus, we have presented the results of the original von Mises distribution.

## Results and discussion

For the sake of brevity and coherence, we have organized the results and discussion according to our three principal research questions.

### 1. Is there evidence of looking-at-nothing during the RI of a visual WM task?

We first assessed whether looking-at-nothing behavior was evident by examining the fixation rate to the memoranda compared to the center or other locations during the encoding and RI phases (see Fig 3 and Table 1). As a reminder, Experiments 1 and 2 used different eye trackers that sampled at different rates (1000 Hz and 150 Hz, respectively), and thus the overall fixation rates between the experiments should not be interpreted. We first assessed looking-at-nothing during encoding and the RI for each individual experiment, and then considered fixations to the cued locations in Experiment 1B.

**Experiments 1A and 1B.**   As explained previously, we separately conducted Phase x RI x Location mixed effects models on fixation rate for Experiments 1A and 1B (see Table 1 left and middle columns and Fig 3A and 3B). Fixation rate toward the dots was greater compared to the center and other locations during both the encoding and RI phases as reflected in the main effect of Location (center vs. dots). This pattern occurred regardless of the RI duration, although the fixation rate to the dots was greater during the 2.5s than 4s RI (estimates = 0.47 [0.23, 0.73] and 0.52 [0.34, 0.71] for both Experiments 1A and 1B, respectively), thus yielding credible three-way interactions (see Fig 3A and 3B, respectively). This is consistent with

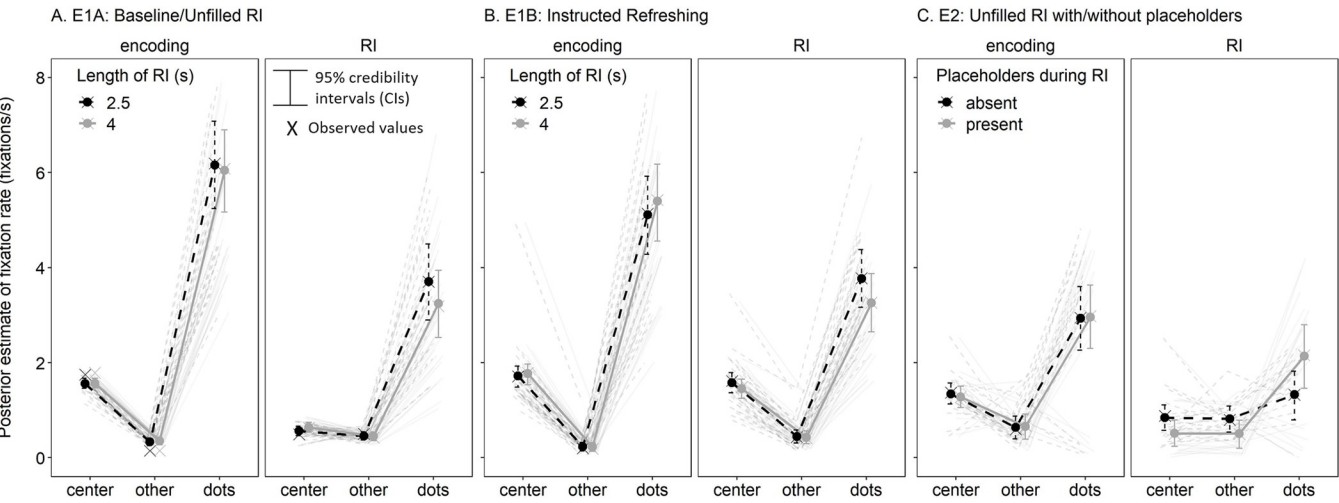

**Fig 3. Research Question 1: Is There Evidence of Looking-at-Nothing During the Retention Interval (RI) of a Visual WM Task?** Fixation Rate (Fixations per Second) Was Highest for Memory Locations (Dots) Compared to Center and Other Locations During Encoding and the RI in Experiments 1A (Panel A) and 1B (Panel B). For Experiment 2, Dot Fixations during the RI Were Only Credibly Higher When Placeholders Were Present (Panel C). The Dark Lines Represent the Posterior Means and the Lighter Individual Lines Represent Individual Participant Means. The Dots represent the Posterior Mean and the Xs Represent the Mean of the Observed Values.

previous findings that fixation rate tends to decrease over the duration of the retention interval (Souza et al., 2020), possibly due to a reduction in novelty.

**Experiment 2.** Given that fixations toward any direction on the screen could be classified as looking at the memoranda when using a circular array as in Experiment 1, Experiment 2 presented the memoranda in only half of the screen to more precisely associate fixations to the memoranda's locations. Furthermore, to assess if gaze support was necessary to observe looking at nothing, placeholders were either absent or present during the RI. Indeed, a credible three-way interaction between Phase, Placeholders, and Location indicated that fixation rate was sensitive to the placeholders during the RI. Fixation rate to the memoranda's locations was only credibly greater than the center (estimate = 1.63 [0.83, 2.41]) and other locations (estimate = 1.64 [0.88, 2.42]) when the placeholders were present during the RI, but no credible differences between locations were observed when placeholders were absent (see Table 1 right column and Fig 3C). Thus, although fixations toward the memoranda showed a similar pattern between the encoding and RI phases in Experiments 1A and 1B, this was not the case in Experiment 2 wherein memoranda were presented in a specific region of the screen unless placeholders were presented. Since Experiments 1A and 1B did not include placeholders and memoranda were presented across all screen, fixations during the RI in Experiments 1A and 1B may largely reflect random gaze rather than looking-at-nothing.

**Experiment 1B.** We also investigated whether looking-at-nothing was sensitive to the instructions to refresh the memoranda in Experiment 1B (see Fig 4). There are two ways to consider this: First, we considered whether fixations to the memoranda during the RI depended on the number of cues to their locations. This allowed us to assess whether participants' fixations were directed to the cued location by the refreshing cues as they appeared during the RI. We counted fixations to the cued locations only if the cue had been presented up to that point during the RI given the potential delay between cue onset and any subsequent change in fixations. Furthermore, we scaled fixation rate according to the number of items in the array that were not cued, cued once, or cued twice for each trial and for each participant given that the task was individually calibrated. That is, given the A-B-C and A-B-A cue

E1B: Instructed Refreshing

**Fig 4. Research Question 1: Is Looking at Nothing Sensitive to Instructions to Refresh the Memoranda in Experiment 1B?** Fixations to the Memoranda Decreased as the Presented Cues to those Locations Within the RI Increased (Panel A). However, Fixations to the Eventually Tested Target (i.e., the Predictor for Research Question 3), Increased as Cues to its Location Increased (Panel B). The Dark Lines Represent the Posterior Means and the Lighter Individual Lines Represent Individual Participant Means. The Xs Represent the Mean of the Observed Values.

sequences, only one item can be cued twice (i.e., A in an A-B-A trial), whereas there could be one (A-B-A) or three (A-B-C) items cued once, and the remaining items of the array are not cued. Thus, scaling fixation rate according to the number of items falling in each cue frequency category corrects for this artifact. We observed a positive effect of cue frequency on fixation rate to the cued locations (estimate = 1.47 [1.03, 1.95]), such that fixations to the cued locations increased with increasing refreshing cues (see Fig 4A). Furthermore, there was a negative effect of cue duration that just overlapped with 0 (estimate = -0.33 [-0.66, 0.00]), such that fixation rate was lower overall for cues presented for 1s compared to 0.5s. The interaction was not credible (estimate = -0.03 [-0.35, 0.29]).

Our second approach examined whether fixations to the eventually tested target increased with the number of times it was cued at any point during the RI. This allowed us to assess whether the fixations used to predict target recall in the next analyses correlate with the number of times the target was cued. Similar to the previous analysis, we observed a positive effect of cue frequency on fixation rate to the target (estimate = 1.64 [1.24, 2.04]), a clearer negative effect of cue duration (estimate = -0.59 [-1.01, -0.20]), and their interaction was not credible (estimate = 0.01 [-0.27, 0.30]; see Fig 4B).

**Conclusions.** The fixation rate analyses indicated that participants most often looked toward the locations of the dots during the RI across experiments, initially suggesting looking-at-nothing behavior. However, the fact that this looking behavior was specific to when the dot placeholders were presented in a specific region of the screen in Experiment 2 suggests that many of the fixations in Experiments 1A and 1B may rather reflect random gaze than looking-at-nothing. Notwithstanding, participants' fixations during the RI of Experiment 1B tended to correlate with the number of refreshing cues to the cued and eventually tested locations. These

results overall suggest that looking-at-nothing in a visual WM paradigm may occur more reliably when there is an on-screen scaffold to support it, such as placeholders (Experiment 1A) or refreshing cues (Experiment 2).

## 2. Does spontaneous looking-at-nothing during the RI of a visual WM task predict recall precision?

We next considered whether spontaneous looking-at-nothing behavior in Experiments 1A and 2 predicted recall precision (i.e., kappa; see Fig 5). Accordingly, we examined the effect of fixation frequency to the target(s) and its interaction with length of the RI (2.5s or 4s; Experiment 1A) or the presence of placeholders (absent or present; Experiment 2). Given that all the items were tested in Experiment 2, test order was also included as a covariate in its analysis.

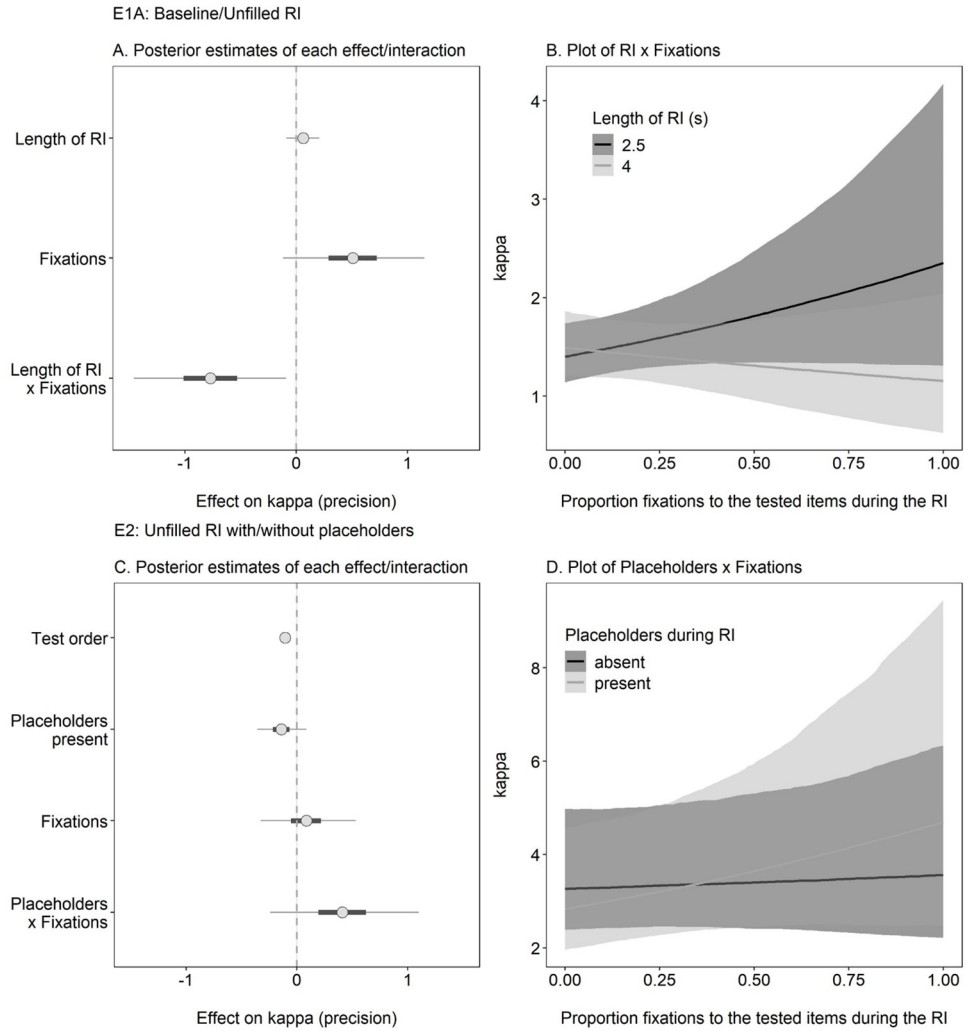

**Fig 5. Research question 2: Does spontaneous looking-at-nothing during the retention interval (RI) predict recall precision? Our results show no credible relation between fixations and recall precision.** Panel A Shows the Posterior Estimates of the Effect of Predictors (Dot = Mean; Thin Line: 95% CI) in Experiment 1A Revealed No Credible Effect of Fixations; Model Predictions (Panel B) Showed No Credible Change in Kappa with Increases in Fixation Proportion or RI Duration. The Posterior Estimates (Panel C) Again Revealed no Credible Effect in Experiment 2; Accordingly, Model Predictions (Panel D) Reveal no Credible Increase in Kappa as Function of Fixation Proportion and Placeholder Presence.

**Experiment 1A.** In Experiment 1A, there was no credible effect of length of RI on precision (estimate = 0.06 [-0.09, 0.21]), congruent with prior work showing no evidence of time-based forgetting in WM for colors [24, 30]. Most importantly, there was no credible effect of fixation frequency (estimate = 0.51 [-0.12, 1.15]), but there was a credible RI x fixations interaction (estimate = -0.77 [-1.46, -0.09]). Despite this credible interaction, follow-up analyses using the emmeans package [31] to assess the slope for each RI showed that the effect of fixations was not credible for either the 2.5s condition (estimate = 0.51 [-0.11, 1.16]) nor the 4s condition (estimate = -0.26 [-0.89, 0.39]). As shown in Fig 5B, the interaction likely occurred due to a crossover of the RI conditions.

**Experiment 2.** In Experiment 2, there was a credible effect of test order (estimate = -0.11 [-0.16, -0.05]), congruent with prior work demonstrating detrimental effects of output interference [32]. There was no credible effect of placeholders (estimate = -0.14 [-0.36, 0.09], and, most importantly, there was again no credible effect of fixations (estimate = 0.09 [-0.33, 0.53]) nor a credible placeholders x fixations interaction (estimate = 0.41 [-0.24, 1.10]). As evident in Fig 5D, kappa was largely similar regardless of the fixations to the tested item or placeholders during the RI.

**Conclusions.** Overall, the fixations toward memory locations observed in Experiment 1A did not predict recall and hence they do not seem functional. In Experiment 2, we included a control to separate random looking around from true looking-at-nothing, which only occurred credibly when placeholders were present (as we showed in the previous analysis to address Research Question 1). Even so, fixations on memoranda were not functional to memory with or without placeholders. Furthermore, the relatively low level of fixations to the eventually tested target in Experiment 1A cannot explain these results given that all of the items were tested in Experiment 2, and still there was no impact of fixations on performance. Thus, these findings collectively suggest that fixations on memory locations during a RI are unlikely to indicate acts of spontaneous refreshing in WM.

## 3. Does looking-at-nothing during instructed refreshing in WM predict recall precision?

Experiment 1B was designed to address our final research question concerning whether looking-at-nothing under instructed refreshing conditions predicted recall precision from WM over and above the previously observed effects of refreshing cue frequency. Accordingly, we once again examined the effect of the proportion of fixation frequency to the target in Experiment 1B, and further considered its interaction with the number of presented refreshing cues (0, 1, or 2) and their duration (0.5s or 1s; see Fig 6).

As evident in Fig 6, we replicated Souza and colleagues' (2015, 2018) prior work showing a beneficial effect of refreshing cues, such that recall precision improved as the number of cues to refresh the tested item increased (estimate = 0.33 [0.17, 0.48]). Interestingly, there was a credible negative effect of cue duration (estimate = -0.16 [-0.28, -0.03]), such that presenting refreshing cues for a longer duration yielded worse precision. We note that this effect is unlikely to be due to time-based decay since the overall retention intervals match the ones used in Experiment 1A, wherein there was no credible effect of RI. Most importantly, there was a credible effect of fixation frequency (estimate = 0.59 [0.11, 1.11]) that interacted with the number of refreshing cues (estimate = -0.39 [-0.71, -0.07]). This interaction hinted at a decreasing sensitivity of kappa to fixation frequency as the number of refreshing cues increased. Follow-up analyses showed that the effect of fixations was credible for 0 refreshing cues (estimate = 0.51 [0.09, 0.96]), smaller but still credible for 1 refreshing cue (estimate = 0.31 [0.02, 0.59]), and not credible for 2 refreshing cues (estimate = 0.11 [-0.23, 0.42]). No remaining interactions were credible.

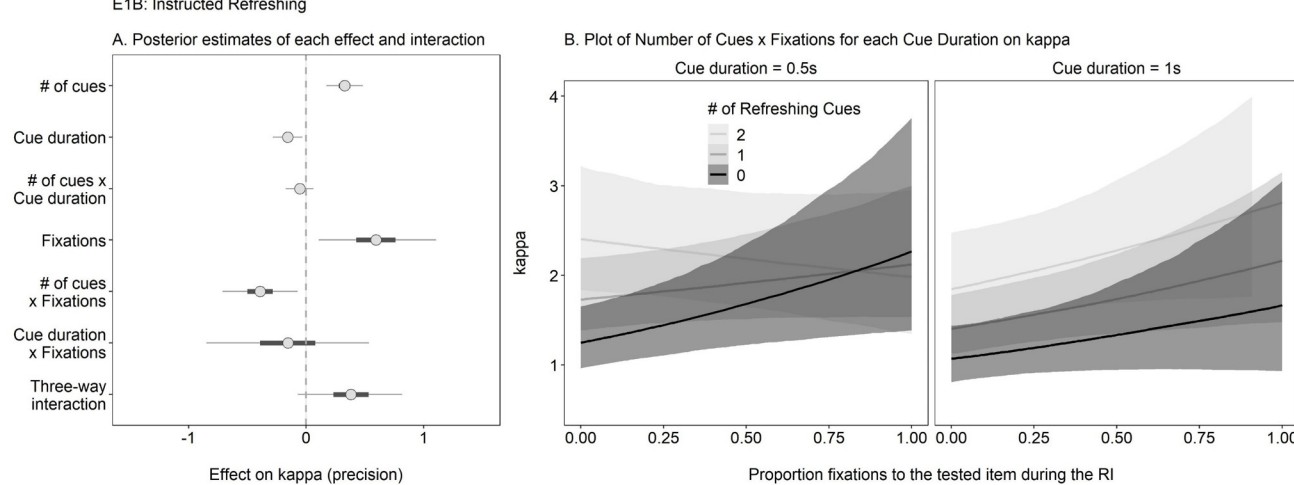

**Fig 6. Research Question 3: Does Looking-at-Nothing During Instructed Refreshing Predict WM Recall Precision in Experiment 1B?** Panel A Shows Posterior Estimates for the Predictors (Dot = Mean; Thin Line = 95% CI) Indicating a Credible Effect of Number of Refreshing Cues, Cue Duration, Fixations, and Interaction of Number of Cues and Fixations. Panel B Presents Model Predictions for the Interaction of Fixation Frequency and Number of Refreshing Cues for Each Cue Duration on Kappa.

In summary, these results indicate that looking-at-nothing behavior under instructed refreshing conditions is functional to WM, and, interestingly, not redundant with the impact of refreshing cues. Fixations appeared to capture additional variance in recall precision that tended to be more pronounced with fewer refreshing cues.

## Overall conclusions

Recording eye movements during the RI of a WM task allowed us to assess if participants looked back to memory locations and whether this looking-at-nothing behavior was functionally related to memory recall, therefore indicating an act of refreshing. Although fixations to memory locations were evident in Experiments 1A and 1B, we could not ascertain how much of these fixations reflected random gaze given that the memoranda appeared in an evenly spaced circular array. Experiment 2 added a control condition: memoranda appeared in only half of the screen. In this case, fixations to memory locations were only credibly higher than to other locations if placeholders were onscreen. Although congruent with prior work showing that placeholders increase fixations [19], these findings challenge the notion that participants purposely look at nothing, particularly when literally nothing is on screen, to refresh the memoranda. Prior work concerning looking at nothing typically includes placeholders [9, 10], but given that most WM paradigms do not, we conclude that fixations under spontaneous settings do not reflect refreshing.

In support of this disconnect between fixations and intention to refresh memory representations, we found no association between fixations and recall precision when participants spontaneously looked at nothing (Experiments 1A and 2). When their attention was cued to refresh the memoranda (Experiment 1B), fixations occurred most frequently to cued locations. This is consistent with prior work showing that retro-cues biased gaze toward cued memory locations [13], and the current work extends this to include refreshing cues that do not reliably indicate the test target as retro-cues do. Thus, fixations corresponded with direct instructions to refresh the memoranda. Fixations to the target also increased with refreshing cues. There was an overall benefit of fixations in Experiment 1B that was of similar magnitude to Experiment 1A, but this time was credible, perhaps because the range of variability was smaller in

Experiment 1B than Experiment 1A. However, the benefit of these fixations decreased as the number of refreshing cues increased, suggesting that fixations may reflect something other than cue use. One possibility is that fixations to non-cued locations indicated some uninstructed refreshing that participants were sneaking in despite the instruction to refresh another item. In that sense, the effect of 0-cued fixations on recall error may be capturing situations in which participants consciously or unconsciously disregarded the cues. Therefore, the previously demonstrated benefit of cue frequency to WM recall [4, 7] cannot be replaced by participants' visual fixations. One caveat is that we used centrally presented cues, and hence cue use could be associated with increases in center-screen fixations. Future studies should consider using peripheral cues to more properly assess the connection between cue use and fixations. Furthermore, the von Mises model that we used only allowed us to assess the effect of fixations on kappa (i.e., precision of recall), whereas there are other established models of visual working memory that include other parameters underlying recall [14, 33, 34]. It is thus possible that fixations under spontaneous or instructed refreshing conditions may affect parameters other than precision, and thus future work will be required to determine this possibility.

Overall, our results suggest that eye movements cannot serve as an online measure of refreshing in WM: Under spontaneous conditions (Experiments 1A and 2), participants did not clearly look at nothing unless placeholders were on screen; but even so, their fixations were not functional to recall. Under instructed refreshing (Experiment 1B), participants tended to look toward cued locations, but the independent benefit of fixations to recall went in the opposite direction of the cues, suggesting that fixations were most beneficial in the relatively rarer instances that participants did not look to the cued locations. These results thus suggest that eye movements are unlikely to indicate acts of refreshing in visual WM, yet may serve as a control to assess cue use in instructed refreshing conditions.

## Acknowledgments

We acknowledge Priyasha Khurana, Maria Sanz Taberner, and Jia Ching See for their assistance with data collection in Experiment 1 and Vanessa Vallesi for her assistance in Experiment 2.

## Author Contributions

**Conceptualization:** Vanessa M. Loaiza, Alessandra S. Souza.

**Data curation:** Vanessa M. Loaiza.

**Formal analysis:** Vanessa M. Loaiza, Alessandra S. Souza.

**Funding acquisition:** Alessandra S. Souza.

**Investigation:** Vanessa M. Loaiza, Alessandra S. Souza.

**Methodology:** Vanessa M. Loaiza, Alessandra S. Souza.

**Project administration:** Vanessa M. Loaiza.

**Resources:** Vanessa M. Loaiza, Alessandra S. Souza.

**Software:** Vanessa M. Loaiza, Alessandra S. Souza.

**Visualization:** Vanessa M. Loaiza.

**Writing – original draft:** Vanessa M. Loaiza, Alessandra S. Souza.

**Writing – review & editing:** Vanessa M. Loaiza, Alessandra S. Souza.

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
