## [Decision Letter · Decision Letter 0]

23 Feb 2022

PONE-D-22-00865Tracking eye movements during spontaneous and instructed refreshingPLOS ONE

Dear Dr. Loaiza,

Thank you for submitting your manuscript to PLOS ONE. After careful consideration, we feel that it has merit but does not fully meet PLOS ONE’s publication criteria as it currently stands. Therefore, we invite you to submit a revised version of the manuscript that addresses the points raised during the review process.

We look forward to receiving your revised manuscript.

Kind regards,

David K Sewell

Academic Editor

PLOS ONE

Journal Requirements:

Dear Dr Loaiza,

Thank you for submitting your manuscript to PLOS ONE (PONE-D-22-00865). I have now received two expert reviews on your submission, and I have also read through it myself. The first review is a joint review signed by Kyle Hardman and Nelson Cowan. The second reviewer has chosen to remain anonymous.

As you’ll see, both reviewers found the manuscript to be clearly written and the studies to be well-designed. I fully agree with their assessments in this regard. Further, the work is on a timely and theoretically interesting topic. There are, however, serious concerns about the interpretation of some of the analyses, which I share. The main issues are as follows:

1 – Conclusions regarding Research Question 3 do not obviously follow from the data. Reviewer 2 highlights the nonmonotonic pattern of effects of fixations on precision (i.e., positive for 1 cue, but null for 2). Reviewer 1 notes there are some issues with coherence of effects across experiments, and identifies apparent discrepancies between the summaries of the analyses and the plots of the effects.

2 – It is sometimes difficult to fully the results due to how this section has been organized around research questions rather than individual experiments. Reviewer 1 notes that some of the analysis details are only implied in places. I agree that more specific information should be included throughout to improve readability. Reviewer 2 comments similarly that it is not always clear which experiment is being referred to with regards to Experiments 1a and 2 (i.e., are they being individuated or lumped together). While I appreciate the purpose of organizing the results as you did, I found it quite difficult to keep abreast of exactly what was being analyzed at any given moment. It may be somewhat less efficient to analyze the study results individually, I think it may be beneficial to assist the reader in following what was done. I am open to keeping the current structure, and it may be the case that more careful attention to detail will solve these issues, but my overall sense is that an experiment-by-experiment structure would be clearer.

3 – There’s a lack of clarity surrounding the data presented in Figure 3. Although you do attempt to address the apparent inconsistency in the results across the two panels in the figure, I didn’t find the explanation especially convincing. Reviewer 1 suggests a potential way to resolve this issue and more meaningfully present the fixation rate results. I think this is an excellent suggestion and encourage you to consider pursuing a “scaled” analysis of these data.

In addition to these three main issues, I add a few smaller comments based on my reading of the manuscript below.

4 – For the ABA trials in Experiment 1b, did the cues always alternate between different items, or could the same item be cued twice in succession? That is, did the “ABA” trials include “AAB” and “BAA” sequences too?

5 – On page 15, you note that responses were centered on the target value. However, no data or analysis are presented to corroborate this. Can you include a plot of the aggregated data and/or could you verify the centering by fitting a von Mises with a freely estimated mean and show (for example) that the credible interval overlaps 0?

6 – The reporting of the analyses for Research Question 1 was ambiguous with regards to what parameter was being affected by the various factors outlined in Table 1. You mention that fixation rate was assumed to be ex-Gaussian distributed. Do the reported estimates and effects pertain to the mean, or are the variance and tau parameters also affected? This needs to be more clearly presented.

7 – This point is mentioned in detail by Reviewer 1, but I had similar thoughts regarding fixation number vs. duration. For example, are fixations to cued locations fewer in number, but longer-lasting? Analysis of fixation duration seems more important than just number of fixations for maters related to attention, as noted by Reviewer 1. There is also some precedent in the categorization literature using fixation duration (or proportion of trial time fixating an item/feature) as an assay of attention (e.g., Rehder & Hoffman, 2005, Cognitive Psychology).

Rehder, B., & Hoffman, A. B. (2005). Eyetracking and selective attention in category learning. Cognitive Psychology, 51(1), 1–41. https://doi.org/10.1016/j.cogpsych.2004.11.001

Overall, I am rejecting the manuscript in its current form, but invite you to submit a revision that addresses (and/or rebuts) all of the points raised by the reviewers. Please take care when preparing your revision, as I would like to avoid a lengthy review process involving multiple rounds of revisions.

I thank the reviewers for their insightful comments, and thank you for submitting your manuscript to PLOS ONE. I very much look forward to receiving your revision.

Yours sincerely,

David K Sewell

Reviewers' comments:

Reviewer's Responses to Questions

**Comments to the Author**

1. Is the manuscript technically sound, and do the data support the conclusions?

Reviewer #1: Partly

Reviewer #2: Partly

2. Has the statistical analysis been performed appropriately and rigorously? 

Reviewer #1: No

Reviewer #2: I Don't Know

3. Have the authors made all data underlying the findings in their manuscript fully available?

Reviewer #1: Yes

Reviewer #2: Yes

4. Is the manuscript presented in an intelligible fashion and written in standard English?

Reviewer #1: Yes

Reviewer #2: Yes

5. Review Comments to the Author

Reviewer #1: In this study, the authors examined whether fixations to memoranda locations during working memory maintenance could be interpreted as reflective of a memory refreshing process. The task was working memory for colors in which continuously-varying colors were presented at differentiated locations. Eye tracking was used during the task to measure eye fixations to memoranda locations.

This study had three main questions and the evidence related to each question is evaluated below. The answers to two questions were reasonable, but there were serious issues with the answer to the third question. In addition, there were some general concerns with the study that are explained in sections below.

Overall, the study appears to be well conducted and the method is suited to answering the research questions. The issues are mostly related to analysis. As such, the recommendation is that the authors submit a major revision that addresses the identified issues.

Regards,

Kyle O Hardman and Nelson Cowan

Research Questions:

Q1: Do participants look at locations of previously presented memoranda? (I.e. the looking-at-nothing effect.)

For Q1, the authors found evidence that participants look at the locations of memoranda during the RI in Exp1A (Fig2). However, in Exp2 when location placeholders were absent, participants had little preference for fixating memoranda locations. In Exp2, fixating on memoranda locations required eye movement to a particular region of the screen, whereas in Exp1A, memoranda locations were in all directions from the initial fixation. Thus, memoranda location fixations in Exp2 are more meaningful than in Exp1A. Based on this evidence, the authors conclusion that participants do not purposefully look-at-nothing is reasonable.

A subcomponent of Q1 is the question of whether cues to memoranda locations can cause fixations (cued looking-at-nothing; Exp1B). It seems like the cues do cause looking-at-nothing (Fig3), but the data in Fig3 could be analyzed better as explained in the section Experiment 1B Fixation Rate Scaling below. Spontaneous looking-at-nothing is a much better indication of a refreshing process than cued looking-at-nothing which may be more reflective of participants doing what they are told to do.

The best answer to Q1 comes from Exp2, which provides a reasonable answer.

Q2: Does spontaneous looking-at-nothing affect recall precision? (Exp1A and Exp2)

For Q2, the authors found evidence against spontaneous looking-at-nothing providing a benefit to recall precision (Fig4). Other than an apparently spurious interaction, there was no clear evidence that looking at memoranda locations provided a benefit. The effect of fixations on precision in Exp1A neared significance, which could explain why other authors find such an effect (discussed more later).

We assume that this analysis examined fixations during the retention interval only, and not fixations during encoding, but we could not find a clear statement to that effect. In revision, adding such a statement would be helpful.

The authors provide a reasonable answer to Q2, but there is the limitation that fixation duration was not included in the analyses (discussed more later).

Q3: Does looking-at-nothing when cued to do so affect recall precision? (Exp1B)

More refreshing cues improved precision (Fig5B), but that is not reflective of looking-at-nothing per se: When cued to a particular memorandum, people may reallocate WM resources to the cued item without fixating the cued location.

More specific to Q3, a main effect of fixations on precision was found and an interaction between the number of cues and fixations on precision was also found (Fig5A and C). There are two issues of different types with these results.

Issue 1: The main effect of fixations in Exp1B goes against the results of Exp1A which found no effect of fixations. The effect in Exp1A is in the same direction as Exp1B and of similar magnitude, but the significance tests go in different directions (Exp1A not significant, Exp1B significant). Maybe there is a true effect of fixations on accuracy that was not quite picked up in Exp1A, which would bring part of the answer to Q2 into doubt (although Exp2 provides the better evidence, so Q2 still has a reasonable answer). The authors might benefit from comparing the Exp1A and Exp1B results in terms of similar magnitudes rather than relying on hypothesis testing logic alone.

Issue 2: The interaction as plotted in Fig5C shows that fixations provide the most benefit to items that were cued less often. However, the figure seems inconsistent with estimates presented in the text (P20): “This interaction hinted at a decreasing sensitivity of kappa to fixation frequency as the number of refreshing cues increased: Follow-up analyses showed that the effect of fixations was not credible for 2 refreshing cues (estimate = 0.06 [-0.20, 0.33]), positive and credible for 1 refreshing cue (estimate = 0.50 [0.11, 0.90]), and positive but not credible for 0 refreshing cues (estimate = 0.27 [-0.22, 0.77]; see Figure 5C).” The sign for 2 cues disagrees between the text (small positive) and figure (small negative). In the text, the magnitude of the effect on fixations on kappa for 1 cue is larger than 0 cues, but the figure shows the reverse (the slope for 0 cues is steeper than 1 cue). The analyses for this interaction need to be double-checked.

The authors do not adequately answer Q3, but it may be possible to address the issues in revision.

Experiment 1B Fixation Rate Scaling

The pattern in Fig3A seems to be that cues cause participants to fixate 0-cued locations, but that doesn’t make sense and doesn’t agree with Fig3B. One way to address this issue is for the fixation rates in Fig3A to be scaled based on the proportion of items that are cued 0, 1, or 2 times. More explanation:

One trial type is ABC and the other ABA. On ABC trials there are three single cued items and N-3 uncued items. On ABA trials there are 1 double cued item, 1 single cued item, and N-2 uncued items. The authors say N is usually 5 or 6 and let’s imagine N=6 for sake of example. On a combination of 1 ABC trial and 1 ABA trial are 12 total items, and in terms of cues a total of 1 double cued, 4 single cued and 7 uncued items. Fixation rate, averaged (by eye) across cue durations are approximately 4 for double cued, 4.5 for single cued, and 5 for uncued. Taking the number of items of each cue type into account, double cued items had a fixation rate of 4 for 1 item (ratio=4), single cued had 4.5 for 4 items (ratio=1.1), and uncued had 5 for 7 items (ratio=0.7). Looking at the ratios shows that participants are fixating cued items at a rate higher than would be expected given the number/proportion of items of each type (vs the assumption of random fixations), but Fig3A seems to suggest the opposite. It is possible to take the proportions of cued item types into account when making Panel A? Or make an additional panel taking proportions into account?

Fig3B seems to show the real effect better because the proportions of cue types (double, single, uncued) are “equalized” (in a sense) because there is only 1 target. Even if there are more uncued items than other types, only 1 of those can be selected as a target, so fixations to other nontarget uncued items are not counted in Panel B.

The authors attempt to explain the discrepancy between Fig3A and Fig3B on P18 in the paragraph starting “At first glance, the positive effect of cue frequency in this second analysis appears to conflict with the negative effect in the first analysis.” Taking the cue type proportions into account is a better explanation than the one given in this paragraph.

Also in the discussion P21: “When their attention was cued to refresh the memoranda (Experiment 1B), fixations occurred most frequently to non-cued locations.” While this statement is true, it is misleading because it doesn’t take the cue type proportions into account.

(To be clear: the analyses of recall precision, like in Fig5, should not be affected by the concerns about fixation rate scaling.)

Analyzing Fixation Duration

No estimate of the duration of a look was provided. The dependent variable for eye movements was “looking rate” defined as fixations per second, but fixation rate doesn’t take into account the total duration of looking at target versus non-target locations. For example, suppose each look to a target location lasts three times as long as each random look elsewhere. That would be an indication of preferential looking at a target location that is not picked up by the looking rate measure. It is not implausible that meaningful looks would last longer than non-meaningful ones.

When it is stated that “fixation rate decreased with increasing cues to memoranda locations (estimate = -1.21 [-1.87,-0.54]), such that participants looked more often to never-cued locations compared to once or twice-cued locations (see Figure 3A)” maybe that counterintuitive finding occurs because participants are looking at cued locations longer with each look.

It is possible that fixation duration is reflective of resource allocation or trying to remember/reconstruct the item at a location. It could be that analyzing fixation duration along with fixation rate could provide a more complete understanding of the data. In revision, the authors should examine fixation duration. If it is a poor measure, the reason why it is a poor measure should be given.

Model choice and purpose of modeling

It isn’t clear what working memory model is being used. It sounds like a precision-only model in that nothing is mentioned about any parameters other than precision (kappa). Is the used model like the Bays and Husain (2008) model?

Given the array sizes that were used, it is possible that many participants were below capacity, so some of the observed effects on precision could have been due not to reduced precision per se, but fewer items being in working memory (e.g. changes in the Pm parameter in the Zhang and Luck 2008 model). Readers could interpret kappa as memory precision alone with forgetting accounted for by other parameters in the model. The authors should address this point of confusion about the model.

It seems like the model is used as a kind of data transformation and that the analyses could be performed on the raw data (degrees of response error) instead of kappa. The authors should provide a reason for using the model instead of raw data.

Bays, P.M., & Husian, M. (2008). Dynamic Shifts of Limited Working Memory Resources in Human Vision. Science, 321, 851-854.

Exp1B Minor Issues

The language Within Trials vs Across Trials in Fig3 is unclear. Presumably, the data are aggregated across all trials for both plots. Also one trial does not affect another trial, so that is not what is meant by across trials.

Exp1B method should clearly state whether the cues were informative (increased probability of testing the cued locations vs random test location). In the discussion random testing is suggested: “… but in the current work the refreshing cues of Experiment 1B did not clearly indicate the test target as retro-cues do.”

About the Exp1B retention interval duration, the authors say on P20: “… presenting refreshing cues for a longer duration yielded worse precision.” Was the RI longer when cues were longer? The method isn’t clear on this point.

Misc Minor Issues

The explanation on P13 the threshold radius for classifying fixations is confusing, possibly because the units are wrong. The confusing quotes: “This threshold was defined as a radius of 142°, which is the radius around 4 dots (…) before touching the area around the next dot …” “The threshold for looking at the center of the screen was a radius of 57.5° …” Do you mean the center was defined as a circle with a radius of some length? If so, a length in degrees would make me think visual angle, but 57 degrees of visual angle would be too big. Are the radii pixels rather than degrees?

For figures showing parameter intervals (like Fig 5 Panel A): Can you make the vertical line at 0 a little thicker so it stands out better?

For Table 1, can you spell out what the factors are in the three-way interaction? The four factors are Phase, RI, Placeholders, and Location. RI is not present in Exp2 and Placeholders are not present in Exp1, so we guess that the 3-way is Phase and Location plus RI (Exp1) or Placeholders (Exp2). Is this right?

Bayes factors are mentioned on P7, but it seems like you use credible intervals and not Bayes factors.

The graphs of fixation rate should include something like “Rate (looks per second)” on the Y axis so the meaning of “rate” is immediately obvious.

Some relevant literature was omitted: e.g., Mall, J.T., Morey, C.C., Wolff, M.J. et al. Visual selective attention is equally functional for individuals with low and high working memory capacity: Evidence from accuracy and eye movements. Atten Percept Psychophys 76, 1998–2014 (2014). https://doi.org/10.3758/s13414-013-0610-2

Further Questions

Was there a tendency for locations fixated during encoding to be fixated again during retention? If this answer is yes: Might the benefit (if any) of fixating a location during retention be an artifact of attention returning to stimuli fixated during encoding, which are presumably encoded better than non-fixated stimuli? Specifically, when modeling memory precision, if fixations to the target at encoding are included as a covariate, does that account for some of the effect of fixations during retention on precision? In Exp1B, can you make the effect of fixations on precision go away by including fixations at encoding as a covariate? These questions might be better answered by researchers who find a substantial looking-at-nothing benefit than the current authors who found some benefit only in Exp1B. There is no need to answer these questions in revision, but we think they are good questions.

Exp2 RI shows much lower preference for fixating on memoranda locations when placeholders absent vs other experiments. Is this effect is due to the memoranda being presented in a smaller segment of the screen in Exp2 than Exp1? Perhaps there is an effect of the eyes wandering away from the center of the screen that accounts for apparent looking-at-nothing in Exp1 that could be examined with Exp2 data. This could be done by dividing the screen into four sections: stimulus locations used on this trial, stimulus locations not used on this trial (i.e. the stimulus ring around the center, not including this trial’s quadrant), center, and other. Presumably, however, stimulus locations not used on this trial would count as “other” in Fig2. If there was an eyes wandering from center effect, “other” should have a higher fixation rate during RI than was observed. So there does not appear to be an eye wandering effect, but where do participants eyes go during retention in Exp2? It seems like fixation rate is just lower in Exp2 than Exp1A and 1B. Maybe this has to do with fixation rate vs fixation duration? There is no need to address these question in revision, but it is a curious finding.

Reviewer #2: This study presents two experiments of which the goal is to verify whether eye movements can be used as an online measure of refreshing. Participants had to remember arrays of n colors. Experiment 1 used spontaneous and instructed refreshing during the retention interval, testing one item in the end. Experiment 2 used trials with and without placeholders to guide fixations during the retention interval, and tested all of the memory items.

The conclusion of the study is less clear to me, I have difficulty giving a clear take home message. On the one hand the authors state that “fixations during instructed but not spontaneous refreshing conditions account for additional variance in recall precision.” But then also:” Eye movements however, do not seem suitable as an online measure of refreshing”.

IF (!) I understood everything correctly, the conclusion should just be that eye movements do not seem suitable as an online measure of refreshing. (see comment below p 20-21) Is that correct?

The abstract is written very clear (except for the conclusion)

The research question is very relevant and immediately captured my interest! And even though the answer may be negative in the end, at least we have an answer to a pending question!

I found it difficult to follow the manuscript at several moment. I wonder if it might be easier to split the experiment in experiment 1 and then Experiment 2, and for the analysis clearly split those for experiment 1A and 1B. I was often confused as Experiment 1A and 2 are often taken together for the analysis but this is not always clearly indicated so this led to confusion for me on several moments (whether the paragraph concerned only Experiment 1A or Experiment 1 as a whole). I think it may be easier to start by Experiment 1, show it’s shortcomings and then state in what way Experiment 2 can answer those.

P 13 There is only 14% of the fixations qualified as looking toward the eventually tested item. It is stated that for that reason Experiment 2 was done. Is there an explicit reason that the instructed refreshing condition is not replicated? As this one seemed potentially to lead to interesting results?

P 16 Is there a reason why the fixation rate would be greater during the 2.5 then during the 4 s interval ( and would the fixation rate be similar during the first 2.5 seconds? And then after 2.5 seconds there would be less fixations)

P 20-21

“Follow-up analyses showed that the effect of fixations was not credible for 2 refreshing cues (estimate = 0.06 [-0.20, 0.33]), positive and credible for 1 refreshing cue (estimate = 0.50 [0.11, 0.90]), and positive but not credible for 0 refreshing cues (estimate = 0.27 [-0.22, 0.77]; see Figure 5C). All the remaining interactions were not credible.

In summary, these results indicate that looking-at-nothing behavior under instructed refreshing conditions is functional to WM, and, interestingly, not redundant with the impact of refreshing cues. Fixations appeared to capture additional variance in recall precision that tended to be more pronounced with fewer refreshing cues.”

I have my doubts about this latter conclusion. The authors say that the effect is positive for 1 refreshing cue. That seems good news and in line with the conclusion that looking- at -nothing behavior under instructed refreshing is functional to WM. However, the fact that the effect was not credible for 2 refreshing cues nullifies this conclusion to me. Having 2 refreshing cues should at least have the same effect as 1 refreshing cue (and then maybe more but not per se). Can the authors elaborate more on their point of view as to why the evidence supports this conclusion?

In general, I guess the manuscript has a potential that is larger than it shows right now if it would be rewritten in a more straightforward way and with a clear conclusion.

6. PLOS authors have the option to publish the peer review history of their article (what does this mean?). If published, this will include your full peer review and any attached files.

Reviewer #1: **Yes: **Kyle O Hardman

Reviewer #2: No

---

## [Author Response · Author response to Decision Letter 0]

27 May 2022

Please see the attached cover letter.

---

## [Decision Letter · Decision Letter 1]

24 Jun 2022

Tracking eye movements during spontaneous and instructed refreshing in working memory

PONE-D-22-00865R1

Dear Dr. Loaiza,

We’re pleased to inform you that your manuscript has been judged scientifically suitable for publication and will be formally accepted for publication once it meets all outstanding technical requirements.

Kind regards,

David K Sewell

Academic Editor

PLOS ONE

Additional Editor Comments (optional):

Dear Dr Loaiza,

Thank you for submitting this careful revision of your manuscript. I have received reviews from the same two reviewers who assessed the original manuscript (Reviewer 1 is Kyle Hardman; Reviewer 2 opted to remain anonymous). Both of the reviewers indicate that all of their concerns have been addressed in your revision. I very much appreciate the care you have taken in revising the manuscript. The manuscript reads very clearly and the refinements to earlier analyses have greatly helped to clarify some of the results that were previously ambiguous.

I do note that Figure 1 appears to be missing from the revised manuscript and urge you to take care when uploading the figures for the final submission.

Setting aside the omitted figure, I am happy to accept the manuscript for publication. Congratulations!

Yours sincerely,

David K Sewell

Reviewers' comments:

Reviewer's Responses to Questions

**Comments to the Author**

1. If the authors have adequately addressed your comments raised in a previous round of review and you feel that this manuscript is now acceptable for publication, you may indicate that here to bypass the “Comments to the Author” section, enter your conflict of interest statement in the “Confidential to Editor” section, and submit your "Accept" recommendation.

Reviewer #1: All comments have been addressed

Reviewer #2: All comments have been addressed

2. Is the manuscript technically sound, and do the data support the conclusions?

Reviewer #1: Yes

Reviewer #2: Yes

3. Has the statistical analysis been performed appropriately and rigorously? 

Reviewer #1: Yes

Reviewer #2: Yes

4. Have the authors made all data underlying the findings in their manuscript fully available?

Reviewer #1: Yes

Reviewer #2: Yes

5. Is the manuscript presented in an intelligible fashion and written in standard English?

Reviewer #1: Yes

Reviewer #2: Yes

6. Review Comments to the Author

Reviewer #1: The authors responded well to our concerns and have much improved the manuscript in this revision. Our concerns have been addressed in this revision and the article seems ready to publish from our perspective. We appreciate the extra figures that were included in the responses to our comments.

Regards,

Kyle O. Hardman

(Nelson Cowan co-reviewed the original manuscript but not this revision.)

Minor notes:

In Figure 4, its possible that panels A and B are not both needed. Cues are uninformative and participants don’t know what item will be tested, so there should not be a systematic difference in fixating target and nontarget items. There could be more noise in panel B because the target is only 1 of many items. You may only need to include panel A.

Where did Figure 1 go? I assume it’s the same as the first submission.

Page 18 last line: “associated” -> “associate”

Reviewer #2: All comments have been adressed appropriately, I have no furher comments.------------------------------------

7. PLOS authors have the option to publish the peer review history of their article (what does this mean?). If published, this will include your full peer review and any attached files.

Reviewer #1: **Yes: **Kyle O. Hardman

Reviewer #2: No

---

## [Editor Report · Acceptance letter]

6 Jul 2022

PONE-D-22-00865R1 

The eyes don’t have it: Eye movements are unlikely to reflect refreshing in working memory 

Dear Dr. Loaiza:

I'm pleased to inform you that your manuscript has been deemed suitable for publication in PLOS ONE. Congratulations! Your manuscript is now with our production department. 

Kind regards, 

on behalf of

Dr. David Keisuke Sewell 

Academic Editor

PLOS ONE